# LANGUAGE-BIASED IMAGE CLASSIFICATION: EVALUATION BASED ON SEMANTIC REPRESENTATIONS

**Yoann Lemesle**[*]
INRIA, France
ENS Rennes, France

**Masataka Sawayama**[†*]
INRIA
France

**Guillermo Valle-Perez**
INRIA
France

**Maxime Adolphe**
INRIA
France

**Hélène Sauzéon**
INRIA, France
Université de Bordeaux, France

**Pierre-Yves Oudeyer**
INRIA, France
Microsoft Research Montreal

## ABSTRACT

Humans show language-biased image recognition for a word-embedded image, known as picture-word interference. Such interference depends on hierarchical semantic categories and reflects that human language processing highly interacts with visual processing. Similar to humans, recent artificial models jointly trained on texts and images, e.g., OpenAI CLIP, show language-biased image classification. Exploring whether the bias leads to interference similar to those observed in humans can contribute to understanding how much the model acquires hierarchical semantic representations from joint learning of language and vision. The present study introduces methodological tools from the cognitive science literature to assess the biases of artificial models. Specifically, we introduce a benchmark task to test whether words superimposed on images can distort the image classification across different category levels and, if it can, whether the perturbation is due to the shared semantic representation between language and vision. Our dataset is a set of word-embedded images and consists of a mixture of natural image datasets and hierarchical word labels with superordinate/basic category levels. Using this benchmark test, we evaluate the CLIP model. We show that presenting words distorts the image classification by the model across different category levels, but the effect does not depend on the semantic relationship between images and embedded words. This suggests that the semantic word representation in the CLIP visual processing is not shared with the image representation, although the word representation strongly dominates for word-embedded images.

## 1 INTRODUCTION

Language is not only a fundamental tool enabling humans to communicate with each other: it also shapes other aspects of perception and cognition, ranging from influencing visual processing to enabling abstract thinking (Vygotsky, 1934; Miller, 1951; Waxman & Markow, 1995; Colas et al., 2021). Analogous to this human ability, recent artificial intelligence technology has also utilized language to drive the training of visual skills in machines. For instance, the OpenAI team introduced CLIP (Contrastive Language–Image Pre-training), consisting of joint learning of language and vision (Radford et al., 2021). The CLIP model efficiently learns visual concepts from natural language supervision and can be applied to various visual tasks in a zero-shot manner. Joint learning of language and vision has also been utilized in the Visual Question Answering (VQA) literature, where artificial models are asked to answer questions about visual content (Antol et al., 2015; Lu et al., 2016; Fukui et al., 2016).

---

[*]equal contribution

[†]masataka.a.sawayama@inria.fr; sawayama@mswym.com

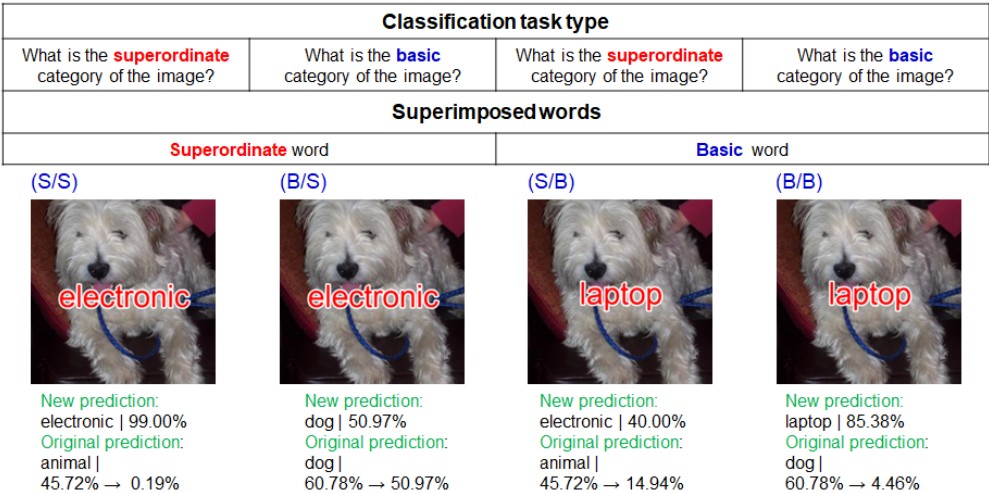

Figure 1: Overview of our benchmark test. Our dataset is a set of word-superimposed images consisting of the natural image datasets and hierarchical word labels. The classification tasks are divided into four conditions: superordinate image classification for the image with the superordinate category word (S/S), superordinate image classification for the image with the basic word (S/B), basic image classification for the image with the superordinate category word (B/S), and basic classification for the image with basic category word (B/B).The classification examples by the CLIP model are shown in Appendix A1.

While joint training with language effectively fosters the acquisition of general abilities for visual recognition/understanding, its abstraction can produce biased recognition for either humans or machines. The Stroop effect known in human cognitive science literature is one of such biases (Stroop, 1935). When human participants observe a mismatched word, where the word "red" is printed with the "blue" color, and are asked to answer the "color" of the word, their response to the word can be slower than when they see the word "red" with the "red" color. In addition to the color-word interference, language also interferes with human image category recognition, known as picture-word interference (Rosinski, 1977; Lupker, 1979). When a participant observes an image coupled with an incongruent category word (e.g., Figure 1), the image categorization speed can be delayed. Similar to the effect in humans, deep learning systems jointly trained on language and vision, e.g., CLIP, show object recognition interference when words are embedded in images (Goh et al., 2021). Specifically, by simply adding a handwritten word to an image, the object recognition can be biased to the superimposed word.

A critical aspect of picture-word interference in humans is that it depends on the semantic relationship between images and superimposed words. For instance, the reaction time on the image classification for word-superimposed images is slower when the image category is semantically similar to the word one (Rosinski, 1977). Based on the accumulated findings, previous works have commonly suggested that at least two different processes mediate the effect in humans, as introduced in detail in Section 2.2. Specifically, when a participant observes a word-superimposed image, an *activation process* synthesizes semantic representations corresponding to the superimposed word and image (Figure 2, activation process). Based on the representations, it has been assumed that a *selection process* decides which possible activation is the answer for the current task.(Figure 2, selection process). The activated representation is shared across visual word forms and images, and the dual activation by words and images confuses the decision in the selection process.

These human cognitive findings raise the question of whether interference in artificial models have some similarities with those observed in humans. Indeed, some artificial models show language-biased image classification, but one possibility is that the bias might emerge irrespective of the semantic interaction between words and images. When we consider a model with text and image encoders for picture-word interference (Figure 2), we can regard the functional role of the image encoder as synthesizing semantic representations for word-superimposed images. Although picture-

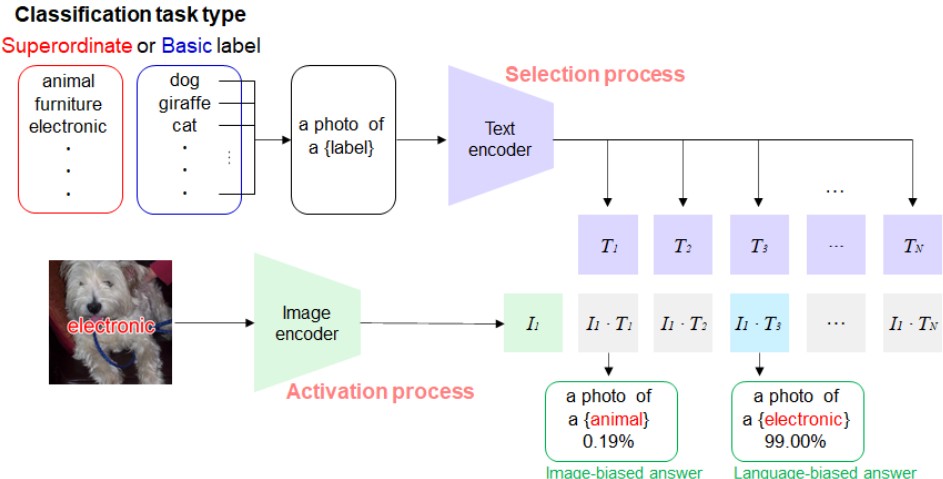

Figure 2: Process diagram of the CLIP model for our task. This figure is created based on Radford et al. (2021). The list of superordinate or basic word labels was used for the classification depending on the task type. We regard the image/text encoder as the selection/activation process in human picture-word interference, as discussed in the main text.

word interference in humans depends on shared semantic representations of written words and images, this is not the only mechanism that can produce the interference. For instance, the image encoder of the model may assign the written word "dog" a different semantic representation than the visual image "dog" and show strong preferential activation to written words while ignoring image contents. Since joint learning of language and vision is expected to enable machines to acquire highly interacted representations of texts and images to solve various general tasks, observing such independent representations and a bias for one of them can be problematic and show a limit of the joint training process.

In this study, we import methodological tools from the cognitive science literature to assess biases of artificial models in image classification for word-superimposed images. Specifically, we introduce a paradigm to evaluate picture-word interference, in which an agent has to predict the image superordinate/basic category for images with superordinate/basic words (Figure 1). Our benchmark dataset is a set of images with superimposed words and consists of a mixture of natural image datasets (Cichy et al., 2016; Mohsenzadeh et al., 2019) and hierarchical word labels (Lin et al., 2014; Krizhevsky et al., 2009). Our benchmark task aims to test artificial models with a text encoder and an image encoder (Figure 2). The task enables us to test 1) whether language-biased decisions happen across superordinate/basic object category levels and 2) the extent to which picture-word interference in artificial models depends on the semantic similarity between superimposed words and images. Furthermore, by importing a methodological tool from the neuroscience literature, called representational similarity analysis (RSA) (Kriegeskorte et al., 2008a), we can also test 3) whether the image encoder has a common semantic representation for superimposed words and images. Out of various joint learning models of language and vision, the present study evaluates the CLIP model using the dataset because the model consists of dual processing of visual and text encoders and is developed to be applied to various general visual tasks without fine-tuning the dataset. To anticipate the result, we show that 1) presenting words disturbs the CLIP image classification even across different category levels, 2) the effect does not depend on the semantic relationship between images and words, and 3) the superimposed word representation in the CLIP image encoder is not shared with the image representation. The present study provides an open-source code to reproduce the results and the full dataset (https://github.com/flowersteam/picture-word-interference).

## 2 RELATED WORKS

**Joint learning of language and vision in artificial models:** Some recent works in machine learning have reported language-biased image recognition for artificial models trained with joint learning of

language and vision. One example is the CLIP model (Radford et al., 2021). The CLIP architecture consists of an image encoder and a text encoder (Figure 2). The image encoder is based on a ResNet (He et al., 2016) or Vision Transformer (Dosovitskiy et al., 2020), and the text encoder is a Language Transformer model (Vaswani et al., 2017). The model is trained on a large dataset of image-text pairs with contrastive objectives, where it learns to align text and image representations for each pair. The pre-trained model can be used for various visual tasks in a zero-shot manner. For instance, when the model is applied for the image classification task, the learned text encoder synthesizes a zero-shot linear classifier for the descriptions, including classification labels, e.g., "a photograph of [label]" (Figure 2). The learned image encoder outputs the representation for an input image, and the text classifier decides which label representation is the closest to the image representation. Language-biased image classification is reported in the CLIP image classification task as a typographic attack (Goh et al., 2021). As shown in Figure 1, when a written word is superimposed on an image, the CLIP classifies the image category as the superimposed word.

Another example of language-biased image classification is reported in the context of the Visual Question Answering (VQA) task (Antol et al., 2015; Lu et al., 2016; Fukui et al., 2016). This task's goal is to answer questions about visual content, e.g., a text question "What color is the dog?" for a dog image. The task also needs joint learning of language and vision, i.e., an image encoder to recognize what the visual content is and a text encoder to understand the question. However, joint learning tends to be biased by superficial correlations between texts and images in the training data and ignore visual input to solve a task (Agrawal et al., 2016). For instance, even when a model sees a green (untypical) banana and is asked about its color, the model tends to answer "the color is yellow" while ignoring the actual image content. Based on the observations, recent VQA models have attempted to remove the biases (Agrawal et al., 2018; Ramakrishnan et al., 2018).

**Picture-word interference in humans:** It is essential to understand how the language-biased processing in artificial models is related to the human mechanisms discussed in the cognitive science literature. This subsection reviews the functional connection between text/image encoders in artificial models and picture-word interference in humans (Figure 2). Picture-word interference refers to language-biased recognition for word-superimposed images, and previous works in cognitive science have investigated this effect using various experimental paradigms (reviewed by Bürki et al. (2020)). Here, we focus on interference when a participant observes a word-superimposed image and answers the image category for the image to compare the underlying cognitive mechanisms with artificial models (Rosinski, 1977).

The discussion about what type of mechanisms mediate picture-word interference is still controversial in the cognitive science literature, especially the stage at which the interference emerges. However, we can summarize common functional mechanisms shared across different theoretical views and ask whether artificial models acquire these fundamental components in the current study. First, most views assume that visual words and images activate common semantic representations, i.e., a written word "dog" activates similar semantic representations to an image "dog." Second, they commonly assume that interference emerges due to some types of selection process. The type of selection process is extensively debated and differs depending on the views. One is the lexical selection hypothesis, and the selection works to compare the semantic representation competitively (Levelt et al., 1999; Roelofs, 1992). This selection relies on relative activation in semantic representations, and picture-word interference emerges because the activation by words induces misselection when it is semantically close to the activation by images. However, the lexical selection is inconsistent with some behavioral findings, e.g., the activation amplitude does not always correspond to the effect size of interference. Another theoretical view assumes the response selection, which is closer to the decision stage than the lexical selection (Finkbeiner & Caramazza, 2006). Since the response selection assumes additional processing for relative activated representations, it can explain the inconsistency of the phenomenon with the lexical selection assumption.

We first ask whether one observes patterns of interference at the behavioral level that are similar to humans and then study more precisely the learned representations to see if the mechanisms proposed to explain human interference could also be at play in the artificial model. To investigate machine mechanisms in terms of the cognitive science findings, we regard an image encoder in the artificial model as the processing to activate semantic representations from images (Figure 2). We also consider a text encoder as the selection process, which tries to find the answers according to the task objectives (superordinate/basic selection).

## 3 LANGUAGE-BIASED IMAGE CLASSIFICATION TEST

Our dataset is a set of word-superimposed images. We use the two image datasets from the cognitive neuroscience literature on object recognition (Cichy et al., 2016; Mohsenzadeh et al., 2019), which include 118 and 156 images with natural object categories. Previous works recorded the spatiotemporal neural activity using fMRI and MEG measurements for the images. The first dataset was annotated with 118 basic categories (Cichy et al., 2016) and the second one with five superordinate categories (faces, bodies, animals, objects, and scenes) (Mohsenzadeh et al., 2019) . For the word labels, we extracted the superordinate category words from the MS-COCO dataset and the basic category words from the MS-COCO and CIFAR-100 datasets (Lin et al., 2014; Krizhevsky et al., 2009). The MS-COCO dataset had the "person" label for superordinate and basic categories. We used this label only for the superordinate category because it was a higher category than other person-related basic labels in CIFAR-100. We removed multi-word labels and duplicates (including the plural of an already existing label) when compounding the MS-COCO and CIFAR-100 basic word label sets. The numbers of superordinate and basic word labels were 12 and 138, respectively. We superimposed all labels on each image, and thus, our word-superimposed image dataset included 41,100 images in total. The color of superimposed words was red and bordered by white lines to enhance the word visibility considering future usage in human experiments. Since the image dataset is supposed to observe objects in the foveal vision, we put the words in the center of the image with a size that largely does not hide the original object.

The classification task was to answer the image label for word-superimposed images. The classification label belonged either to the superordinate or the basic category. The set of labels in the superordinate and basic categories were the same as the words used to make the word-superimposed images. Therefore, the stimulus conditions were divided into four, corresponding to two classification types and two superimposed word categories (Figure 1). The former index S of "S/S" in Figure 1 indicates the classification task level (superordinate or basic), and the latter shows the category level of the word superimposed on the image (superordinate or basic). In addition to these conditions, we used no-word images to evaluate the original prediction by the model.

## 4 EXPERIMENTS AND RESULTS

### 4.1 EXPERIMENTAL SETUP

Performing image classification with the pre-trained CLIP model needs the input of a list of candidate classification labels (or a textual representation of them) for the text encoder and an image for the image encoder (Figure 2). We used the sentence of "a photo of a [label]" for the text encoder. The two encoders output the representations of texts and the image respectively . We then computed the similarities of text and image representations for a set of labels, passed them through the softmax function to obtain the probability of each label for a given input image, and extracted the label with the highest value for the final classification (Radford et al., 2021). We present the results of the pre-trained CLIP image encoder using a Vision Transformer (ViT-B/32) in the main text and those of other image encoders in Appendix C.

### 4.2 LABEL SWITCHING RATE

To test whether superimposing the word affects the classification, we first evaluated the label switching rate. We defined a label-switched image as a word-superimposed image with a different image classification than the corresponding no-word image. The label switching rate refers to the ratio between the number of label-switched images and the total number of images under each classification/superimposed-word condition.

Table 1 shows the results of the label switching rates for each condition. When the classification task type is consistent with the superimposed word category (i.e., the S/S and B/B conditions), the choices of the classification labels include the same word as that superimposed on the image. For these conditions, the CLIP model switched the original classification with high rates. These findings suggest that the model can detect the visual word form on the image and match it with the classification label. These are consistent with the previous work reporting the language-biased image classification of the CLIP model (Goh et al., 2021).

|  | Superordinate prediction | Basic prediction |
|---|---|---|
| Superordinate word | (S/S) 92.77 % | (B/S) 41.24 % |
| Basic word | (S/B) 28.29 % | (B/B) 73.97 % |

Table 1: CLIP's label switching rate for word-superimposed images.

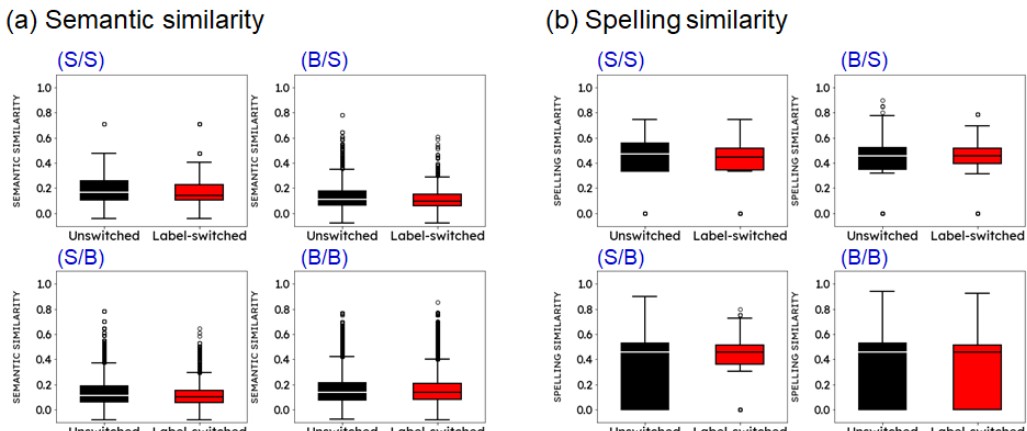

Figure 3: Results of the (a) semantic and (b) spelling similarity analyses.

In contrast, when the classification type is different from the superimposed word category (i.e., the S/B and B/S conditions), the classification labels do not include the same word as the superimposed one. Even under the conditions, the CLIP model showed label-switching for some of the word-superimposed images. Furthermore, some of the switched new labels were related to the superimposed words (Appendix G). The findings suggest that the model can match the meaning of the written word in the image with the label category across different levels. However, it remains unclear whether the image encoder acquires a common semantic representation for the images and the visual word forms. There is also a possibility that the superimposed words are separately represented from the images in the image encoder. (e.g., a superimposed word "dog" is differently represented from an image of a dog). Even so, the label switching across different categories can emerge when the label activation of the text encoder can match with the representation of the superimposed word. To understand the semantic relationship, we evaluated the following semantic similarity between images and superimposed words for the label-switched and unswitched images.

## 4.3    SEMANTIC SIMILARITY

We evaluated the semantic similarity between the original prediction labels for no-word images and the superimposed words to test the semantic dependency in picture-word interference. Specifically, we calculated the vector representations of these labels using a pre-trained Word2Vec model (Mikolov et al., 2013b;a), and then measured the cosine similarity between the two vector representations. If picture-word interference in CLIP depends on the semantic similarity between superimposed words and images, we would expect the similarity in the label-switched condition would be higher than that in the unswitched condition. Figure 3a shows the semantic similarity distribution for the label-switched and unswitched images in the CLIP model. The results show that for all task type and word category conditions, the semantic similarity distributions on the label-switched condition were similar to those on the unswitched condition. This finding suggests that picture-word interference in the CLIP model does not depend on the semantic relationship between images and superimposed words, suggesting that the CLIP image encoder does not acquire a common semantic representation across superimposed words and images.

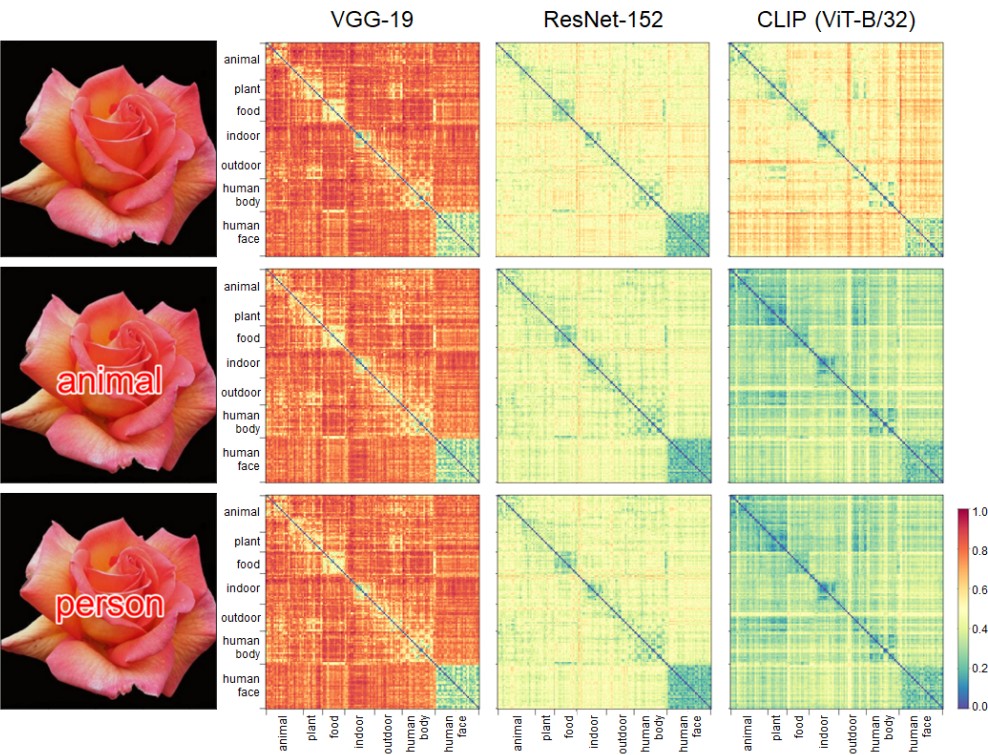

Figure 4: Results of the representational similarity analysis. See also Appendices B1 and H1.

## 4.4 SPELLING SIMILARITY

In addition to the semantic similarity, we evaluated the spelling similarity using the Jaro-Winkler metric (Winkler, 1990) to test how the superimposed word is related to the image category. If the CLIP image encoder acquires a common representation of superimposed words and images based on the edit distance, this metric should change with the label-switched/unswitched condition. Figure 3b shows the result of the CLIP model on the spelling similarity for the label-switched and unswitched images. As in the semantic similarity, the distribution median for the task conditions was always similar between the label-switched and unswitched conditions.

## 4.5 REPRESENTATIONAL SIMILARITY ANALYSIS (RSA)

We also used a methodological tool from the cognitive neuroscience literature, called representational similarity analysis (RSA) (Kriegeskorte et al., 2008a), to understand what is represented in the image encoder for word-superimposed images. RSA refers to an image-by-image similarity assessment of intermediate representations in a brain or model. By computing representational dissimilarity matrices (RDMs) for no-word images and word-superimposed images, we can evaluate how the representation for the no-word images in the CLIP image encoder is affected by adding words on the image. We input one of the two datasets (156 images) (Mohsenzadeh et al., 2019) for this analysis because it consisted of the previously defined superordinate categories. In addition to the CLIP model, we evaluated the two feed-forward convolutional networks (i.e., VGG-19 and ResNet-152) trained for object category classification using ImageNet without joint learning of language and vision to confirm whether presenting words as image manipulation can affect the image encoder without joint learning.

Figure 4 shows the results of the RSA. Each matrix indicates the image-by-image cosine dissimilarity between intermediate features for the paired images. For the features, we used the last fully connected layer for VGG-19/ResNet-152 and the visual encoder output for the CLIP model. The top, middle, and bottom rows show the results for the original images, the word-superimposed images

| prompt | S/S | B/S | S/B | B/B |
|---|---|---|---|---|
| "a red word label over a picture of a X" | 87.69 % | 44.16 % | 29.64 % | 66.23 % |
| "a word is printed in a red font over a picture of a X" | 91.34 % | 44.86 % | 35.29 % | 69.72 % |
| "a photo of a word written in a red font over a picture of a X" | 86.53 % | 43.07 % | 30.78 % | 62.69 % |
| "a text that says X" | 96.78 % | 52.50 % | 54.58 % | 88.22 % |
| "a word of a X is printed in a red font over a picture" | 95.48 % | 51.65 % | 46.28 % | 78.02 % |
| "a photo of the word X written in a red font over a picture" | 96.22 % | 53.58 % | 42.91 % | 85.05 % |
| "a photo of the word Y written in a red font over a picture of a X" | 34.34 % | 35.33 % | 31.04 % | 34.92 % |

Table 2: CLIP's label switching rate for word-superimposed images on different prompts. The red-colored texts are the prompts that specify that the image content is what should be predicted. The blue-colored texts are the prompts that specify that the superimposed word is what should be predicted. The X means the label to be answered by the model. The orange-colored text is the condition in which a prompt varies according to the superimposed word on each image.

with the "animal" label, and the word-superimposed images with the "person" label, respectively. Each left image is one example in the 156 images. The results of other word-embeddings are shown in Appendices B1 and H1.

For VGG-19 and ResNet-152, we observed some representational clusters according to the superordinate categories (i.e., the squared clusters around the diagonal line in each panel with lower dissimilarity values) (Figure 4, top, VGG-19 and ResNet-152). It has been known that higher cortical areas in humans and monkeys also show such categorical clusters for RDMs of natural scene images (Kriegeskorte et al., 2008b; Cichy et al., 2014). Similar to these feed-forward models, the CLIP image encoder also showed representational clusters depending on the superordinate categories (Figure 4, top, CLIP), suggesting that the encoder has a hierarchical structure for visual images.

In contrast, the results of the CLIP model for word-superimposed images showed different representations from that of original images (Figure 4, middle and bottom, CLIP). First, the dissimilarity values for word-superimposed images were generally lower than those for original images (i.e., the representations of all images are similar to each other). The results are consistent with the strong language bias of the CLIP model. As in the other analyses, presenting words on the image shift the CLIP image classification to the direction of the superimposed word. This means that when we put a single word on the different images, their representations can be similar.

Most importantly, our analysis showed that adding words to images did not produce categorically selective effects on the representation of the image encoder. Specifically, we observed category-related squares similar to the original one around the diagonal line of each representational matrix for word-superimposed images (Figure 4, middle and bottom, CLIP). In addition, even when we put different word labels, "person" and "animal," these representational matrices remained similar. If the superimposed words were similarly represented in the image encoder to the image category, presenting words on the image should selectively affect the original image category representation and depend on what words were superimposed. The current results are inconsistent with the hypothesis and suggest that the CLIP image encoder has different representations of word forms (i.e., superimposed words) from visual image representations.

## 4.6 PROMPT DEPENDENCY

The CLIP model is not trained for image classification but just for matching between images and texts. One may consider that the language-biased classification emerges because the task objective for the model, which is defined by the prompt "a photo of X," is ambiguous. The prompt can be both "a photo of the written word" and "a photo of the image content." We tested various prompts to explore the effect of prompts. First, we created three prompts that specify that the image content is what should be predicted by the CLIP model (red in Table 2) and three prompts that specify that the superimposed word is what should be predicted by the CLIP model (blue in Table 2). The label switching rate refers to the ratio of switching word-superimposed images with a different image classification than the corresponding no-word images. In addition to these fixed prompts, we created a prompt that varies according to the superimposed word on each image. We used a prompt "a photo of the word Y written in a red font over a picture of a X," where X is the label to be answered by

the model and Y is the same variable word as superimposed on each image (orange in Table 2). For instance, we used a prompt "a photo of the word electric in a red font over a picture of a X" when the superimposed word is "electric," and a prompt "a photo of the word dog in a red font over a picture of a X" when the superimposed word is "dog."

Table 2 shows the results of the label-switching rate for different prompts. The results showed that the rate changed with the different prompts, i.e., the focus of the task (image content or superimposed word). When the prompt was focused on the image content (red in Table 2) the switching rates were lower than when it was focused on the superimposed word (blue in Table 2). However, the language-biased classification was high for all conditions. The results on the variable prompt (orange in Table 2) showed that the label switching rate for the S/S and B/B conditions decreased more than the prompt conditions that specify the image content (red in Table 2). The finding suggests that describing the superimposed word in the prompt is effective in decreasing the language-biased classification. Even so, the label-switching rates were still high, more than 30% for all conditions. In addition, we analyzed the semantic similarity between the image and superimposed words for label-switched and unswitched images. The results showed that, in any prompts, the semantic relationship did not explain the label-switching effect (Appendix Figure D1).

## 5 DISCUSSION AND CONCLUSION

The present study developed a benchmark test for language-supervised visual models to evaluate the language biases in image classification. Experiments we presented show that adding printed words over images leads to high classification switching rates across different categories of superimposed words and different classification tasks. The finding suggests that the CLIP image encoder can recognize the visual word form as a word, and the model can match the visual word representation with the classification label representations across different category levels. However, the semantic similarity analysis showed that the language-biased image classification in the CLIP model was irrespective of the semantic/spelling relationship between image and superimposed word categories. This finding is different from human findings and suggests that the CLIP image encoder does not acquire the semantic word representation shared with the image representation. The RSA results supported the hypothesis by showing that presenting words on images did not affect the no-word image representation selectively to specific categories.

We consider multiple steps for an artificial model with joint learning of language and vision to acquire semantic compositionality for visual word and image categories. First, an image encoder needs to recognize the visual word form as a word. Next, the semantic representation for words and images needs to be shared in the image encoder. That is, the written word "dog" needs to have the same semantic representation as the image "dog." The present study showed that the CLIP model acquires the first step, but the second step remains challenging. Our language-biased image classification test enables researchers to evaluate step-by-step semantic interactions in joint learning of language and vision.

A remaining question is how the CLIP model acquires the biased representations in the image encoder. The presented results may be improved by further studying them as a function of potential biases in the training data of CLIP itself: for example, it would be useful to understand in the first place how CLIP acquired the ability to visually recognize superimposed words and why we observed a bias favoring the superimposed words over the whole image. In particular, we can ask whether the written words are in the training data and if they are, whether there are adversarial examples in the training data like our word-superimposed images.

Picture-word interference in humans is a phenomenon showing a language-biased effect in terms of image classification. However, this can also be considered as a developmental product humans acquire, where such bias could have functionalities. Some human studies have shown that picture-word interference is inextricably linked to facilitation effects of language priming for image category recognition (Finkbeiner & Caramazza, 2006). The finding reminds us that acquiring human-like biased representations for artificial models can have a functional meaning to process world information efficiently in natural environments. The current efforts to manage language-biased image recognition in artificial models are mainly aimed to remove the biases (Agrawal et al., 2018; Ramakrishnan et al., 2018). However, one direction might be to make the biases more human-like by controlling joint learning curricula of language and vision (c.f., Sigaud et al. (2021)).

## 6 REPRODUCIBILITY STATEMENT

All the source codes are available at our GitHub repository (`https://github.com/flowersteam/picture-word-interference`) to create the word-superimposed images from the image datasets (Cichy et al., 2016; Mohsenzadeh et al., 2019) and to reproduce the current analysis results, which are also shared in the supplementary materials of the ICLR submission. The analysis relies on the pre-trained models shared in the previous studies (e.g., CLIP).

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

## A    PREDICTION EXAMPLE FOR WORD-SUPERIMPOSED IMAGES.

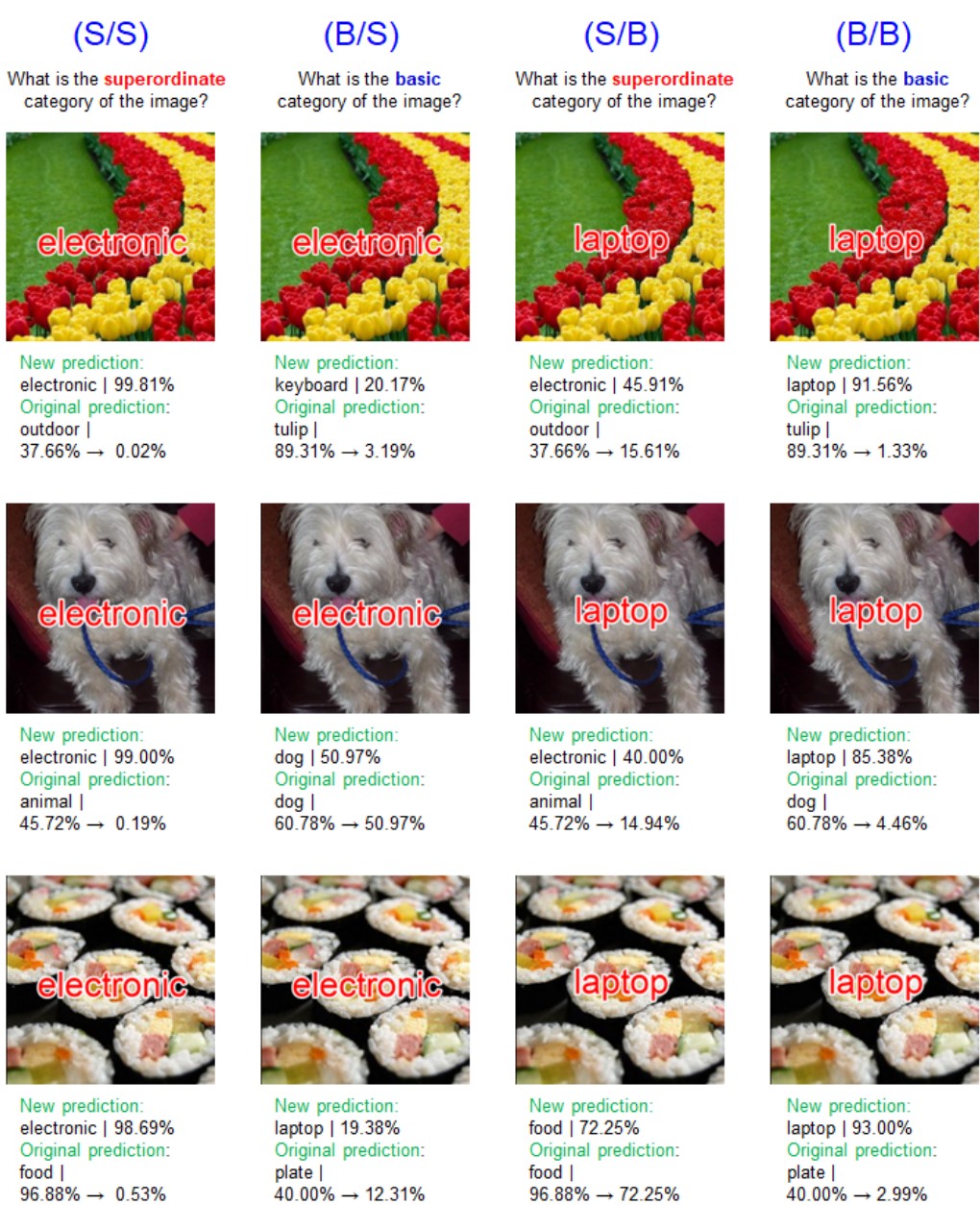

Figure A1:   Prediction example for word-superimposed images using the CLIP model (ViT-B/32). For each stimulus condition (S/S, B/S, S/B, and B/B in Figure 1), the "new prediction" indicates the image classification probability for the word-superimposed image. The original prediction means the image classification probability for no-word images. We also show the extent to which the original prediction changed with the word attachment.

## B  REPRESENTATIONAL SIMILARITY ANALYSIS USING WORD-SUPERIMPOSED IMAGES.

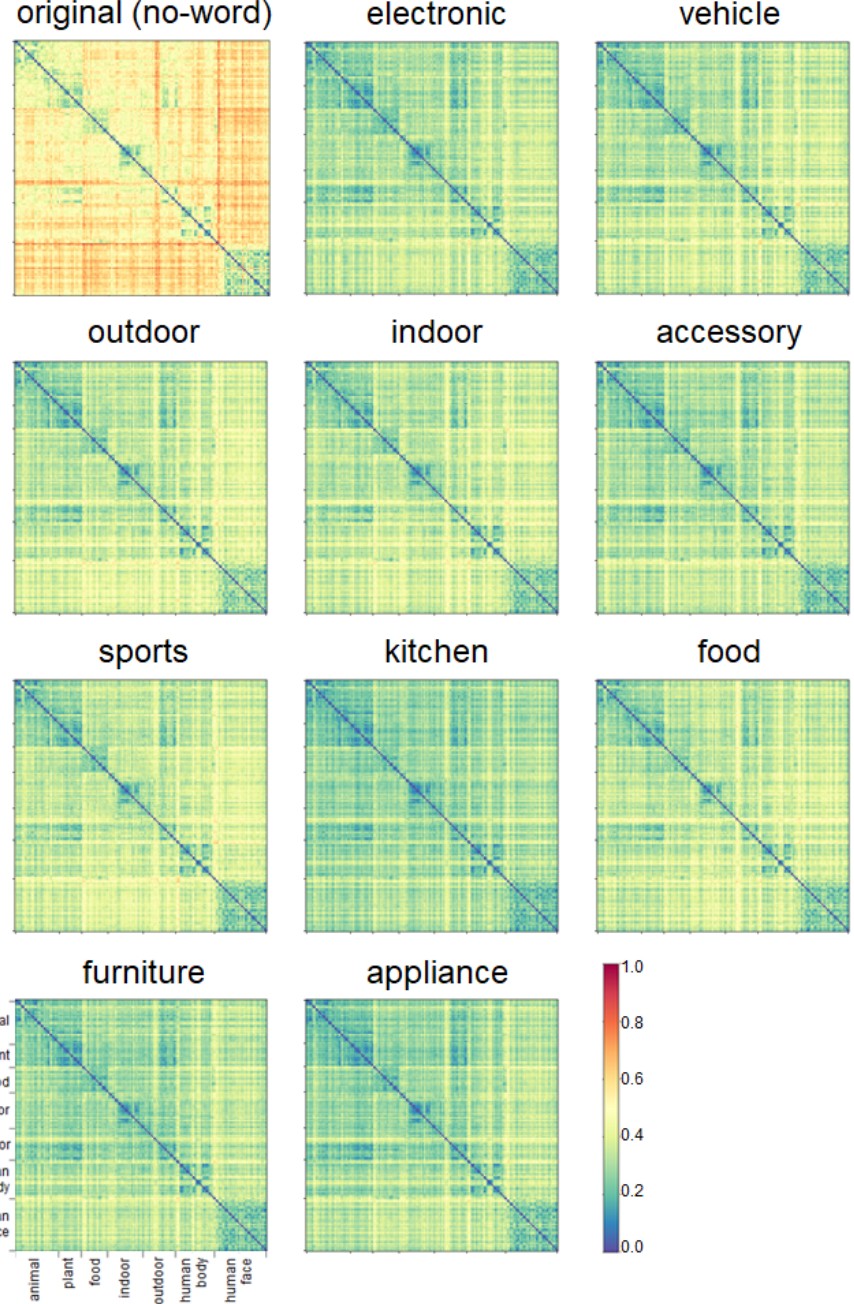

Figure B1: Results of the RSA. In addition to the labels "animal" and "animal" in Figure 4, we show the results of other words. Based on the CLIP image encoder (ViT-B/32) outputs, we calculated the image-by-image representation dissimilarity matrices. Similar to the results of the word labels "animal" and "person," presenting words on the images did not affect the no-word representation in a category-selective way. These results are consistent with the finding in the main text, suggesting that semantic word representations in the CLIP image encoder are different from image representations.

## C    EVALUATION OF DIFFERENT IMAGE ENCODERS OF THE CLIP MODEL

The image encoder of the CLIP pre-trained model is based on ResNets (He et al., 2016) or Vision Transformers (Vaswani et al., 2017). To explore the extent to which the language-biased image classification depends on the network architecture, we investigated the misclassification rate and the semantic relationship using various encoders in addition to a Vision Transformer model (ViT-B/32) used in the main text. Specifically, for the ResNets, we used a ResNet-50, a ResNet-101, and two models which follow EfficientNet-style model scaling (a ResNet 50x4 and a ResNet 50x16). For the Vision Transformer, we used a ViT-B/16 in addition to a ViT-B/32. We used the pre-trained CLIP networks using these architectures, which are shared by the previous work (Radford et al., 2021). The experimental procedure is the same as that of the main text.

Figure C1 shows the results of misclassification rates for the CLIP models using various image encoders. Even when the other architectures were used for the image encoder training, the results were consistent with the performance of the ViT-B/32 version. The misclassification rate was high when the classification type was consistent with the superimposed word level (S/S and B/B). In addition, the CLIP models showed significant misclassification across different category conditions under the other conditions (S/B and B/S).

The results of semantic similarity analysis and the RDA also did not depend on the architecture types (Figures C2 and C3, respectively). The semantic similarity analysis showed the median similarity of the label-switched images was not different from that of the unswitched images Figures C2 for all architectures. The RDA showed that the image encoder representation for no-word images changed with the architectures (Figures C3, first and fourth rows). However, the effect of attaching words did not change with the architecture. For all architectures, the overall dissimilarity across image-by-image pairs decreased for word-superimposed images, but the representation displacements were not selective to the original image representation.

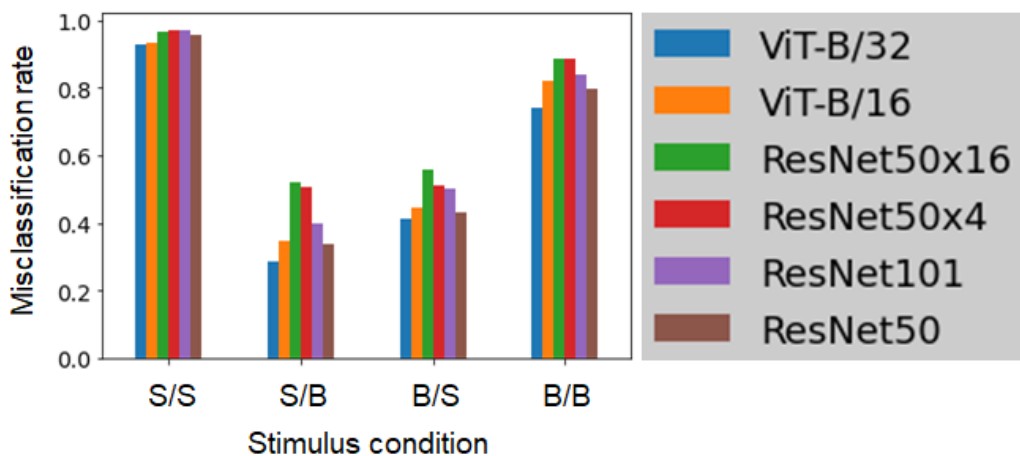

Figure C1:   Results of misclassification rates for the CLIP models using various image encoders. The misclassification rate was plotted according to the stimulus condition (S/S, S/B, B/S, or B/B). Different color labels indicate different architectures.

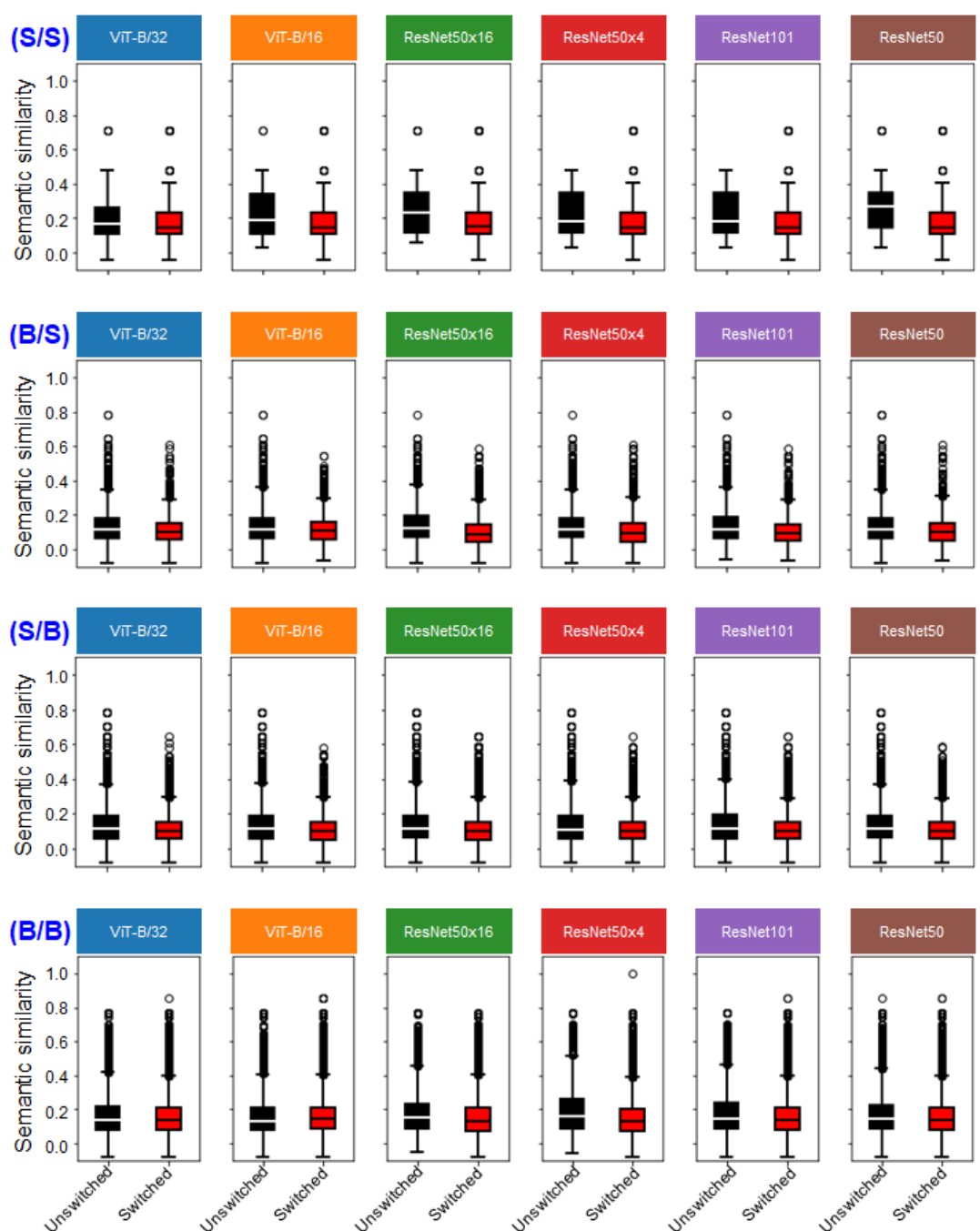

Figure C2: Results of semantic similarity analysis for the CLIP model using various image encoders. The semantic similarity analysis of Figure 3 was conducted for the CLIP models with the image encoder of a ViT-B/32, ViT-B/16, ResNet-50x16, ResNet-50x4, ResNet-101, and ResNet-59. The results of the ViT-B/32 image encoder were replotted from Figure 3. Different color labels indicate different architectures. Each row shows the stimulus condition (S/S, B/S, S/B, or B/B).

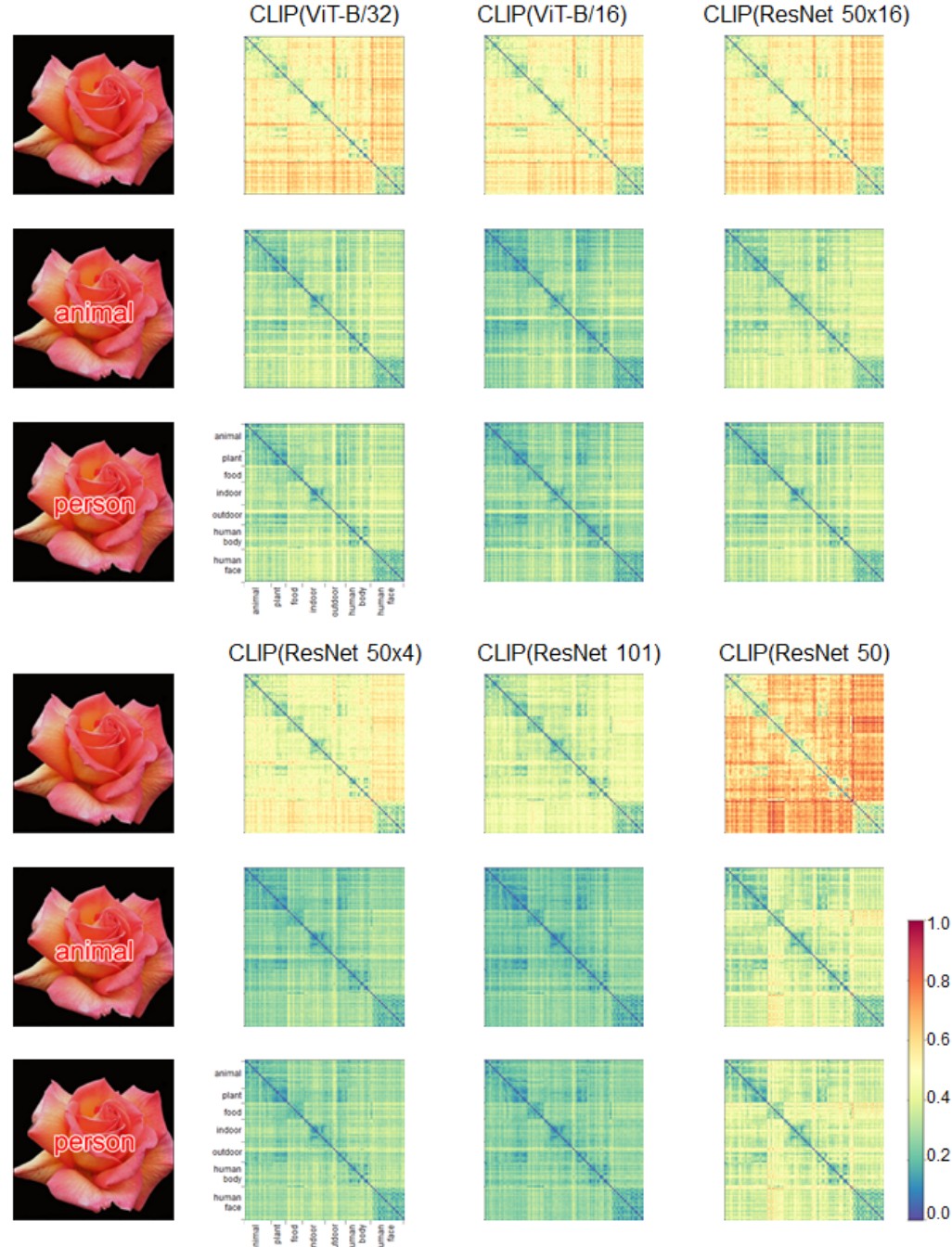

Figure C3: Results of the representational similarity analysis (RSA) for the CLIP model using various image encoders. The analysis method was the same as Figure 4, except that we also used the CLIP models with other image encoders than a ViT-B/32. The results of the ViT-B/32 image encoder were replotted from Figure 4.

# D  PROMPT DEPENDENCY OF LANGUAGE-BIASED IMAGE CLASSIFICATION

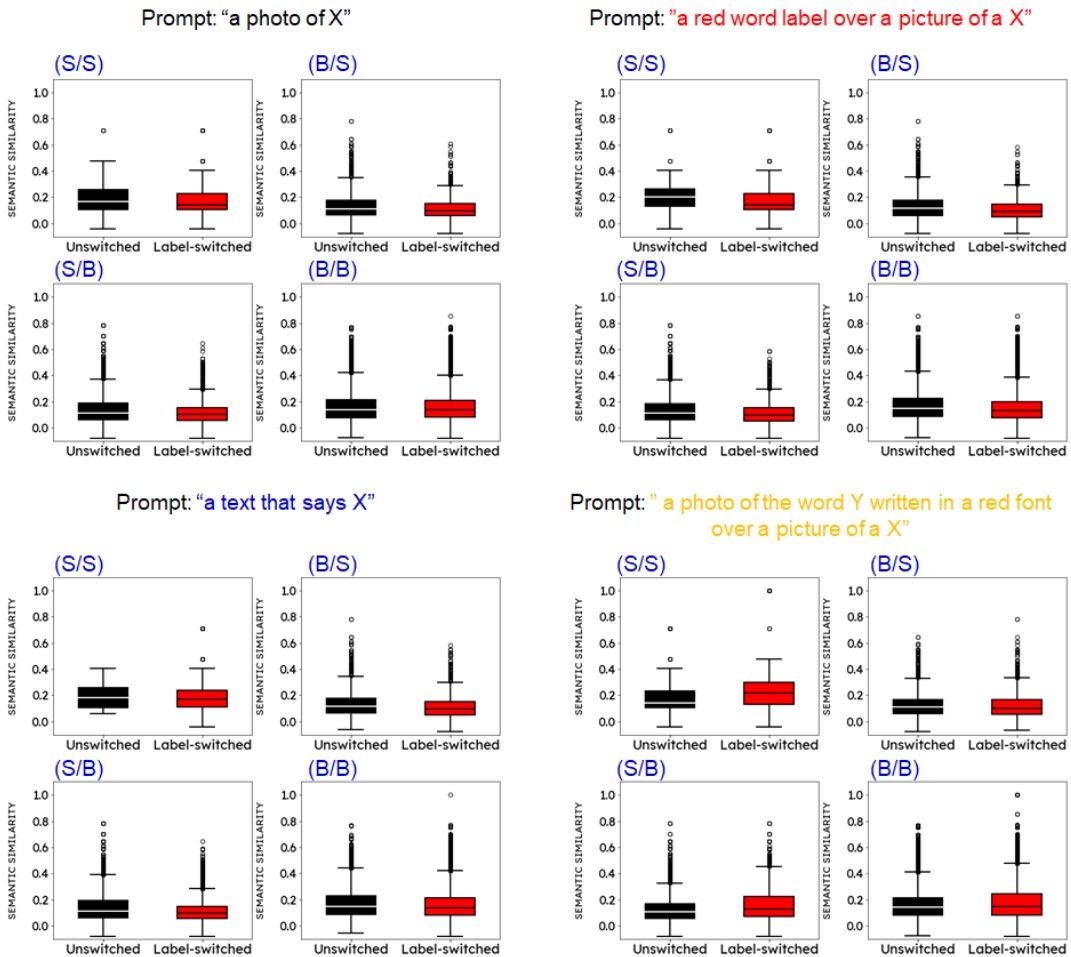

Figure D1:  Results of semantic similarity analysis.The same analysis as Fig was conducted for different prompts shown in Table G1.

## E   PREDICTION PROBABILITY

The prediction probability analysis indicates how label switching relies on the prediction confidence (i.e., the probability of the classified label in the label list) for no-word images and how the original prediction confidence decreases by presenting words. Figure E1 shows the results of the prediction probability analysis. Different panels show different stimulus conditions, as indicated in Figure 1. The original prediction means the probability for no-word images. In the figure, the original prediction is categorized into label-switched or unswitched conditions. The label-switched and unswitched conditions refer to if the original prediction is switched in the new prediction for word-superimposed images. The new prediction shows the probability of the label before and after adding the superimposed word, for the label-switched word-superimposed images.

In the original prediction, the probabilities for the label-switched images were lower than those of the unswitched images. The trend was notable when the task type condition was inconsistent with the superimposed word category (the B/S and S/B conditions). These results suggest that picture-word interference is more effective when the original prediction is not confident. In addition, the original prediction probability drastically decreased in the new prediction for the word-superimposed image (two red plots in each panel of Figure E1). However, we must be aware that a label prediction probability depends on other label predictions and that it can be lower if the other label prediction is high. We show how the original image representation is changed/unchanged by presenting words in the next section. Furthermore, the prediction probability analysis showed that the new label confidence for the label-switched images (the blue plot in Figure E1) was high when the classification task type was consistent with the superimposed word category (the S/S and B/B conditions). This trend is consistent with the results of the label switching rate analysis.

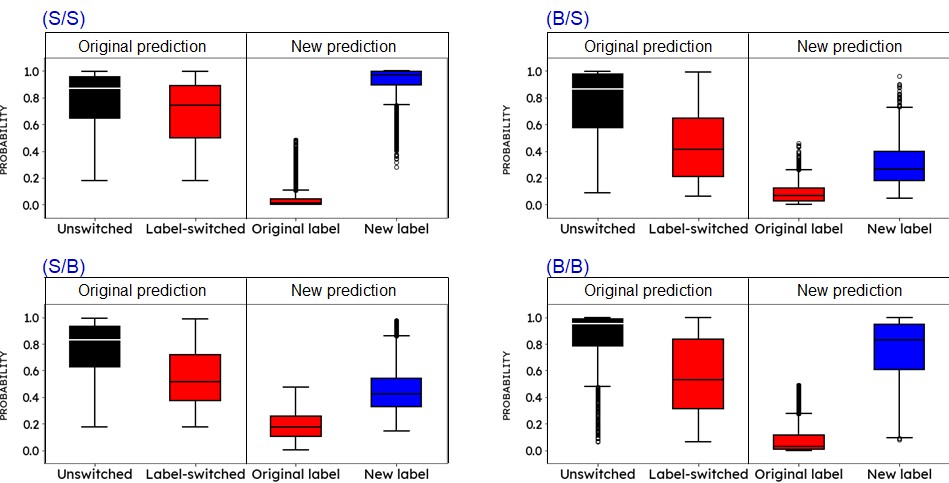

Figure E1: Results of the prediction probability analysis.

## F EFFECT OF PSEUDOWORD

|  | animal (control) | atipol | alipil | mzwxnmsczp |
|---|---|---|---|---|
| Label switching rates | 57.66 % | 26.64 % | 28.46 % | 18.25 % |

Table F1: Label switching rate for nonsense pseudowords.

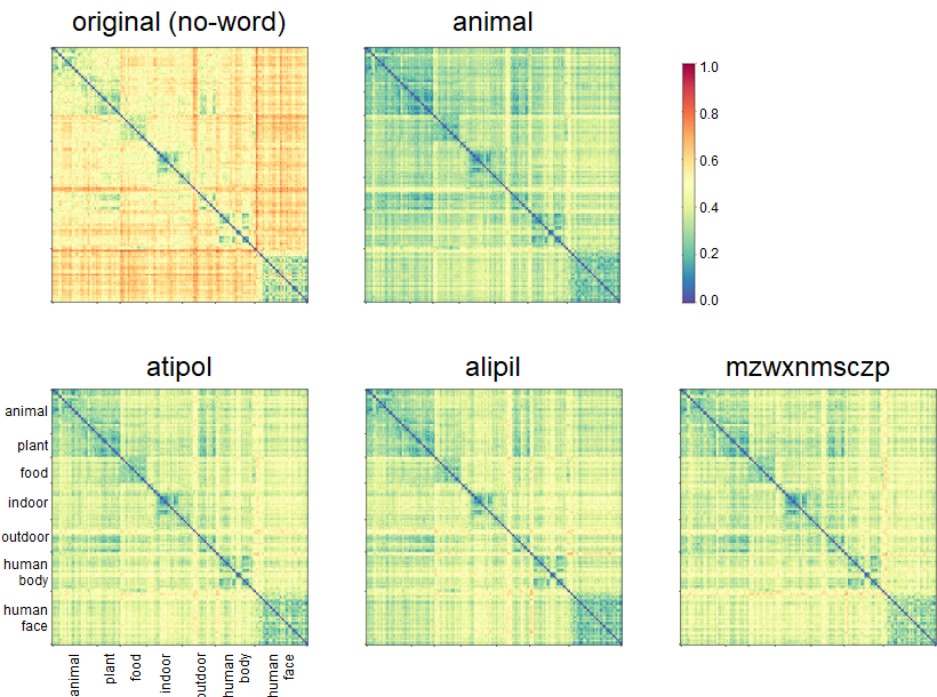

Figure F1: Results of the representational analysis for nonsense pseudowords.

We conducted an additional experiment using nonsense pseudowords and random words. We created two pseudowords from the word "animal" by a multilingual pseudoword generator, Wuggy (Keuleers & Brysbaert, 2010). We also used one random word used in a previous human study (Finkbeiner & Caramazza, 2006). Results showed that the label switching rate decreased for these words (Table G1). We also analyzed the RDM for the images on which these words were superimposed. Our results showed that when nonsense pseudo and random words were superimposed on the images, the effect of words on the representation was weaker than meaningful words. The finding suggests that presenting meaningful words on the images can be a factor affecting the representation of the image encoder.

# G  RELATIONSHIP BETWEEN SWITCHED LABEL PREDICTION AND SUPERIMPOSED WORDS

|  | S/S | B/S | S/B | B/B |
|---|---|---|---|---|
| Consistency rates | 99.28 % | 45.70 % | 36.30 % | 93.49 % |

Table G1: Consistency rates between the switched labels and the superimposed words.We calculated the ratio that the switched labels and the superimposed words were consistent in the label-switched image. This table results from the prompt "a photo of X" using the ViT/32 image encoder. When the prediction category class is same as the superimposed word class (the S/S and B/B conditions), most switched labels were the same word as superimposed on the images (more than 90%) (e.g., a "dog" prediction for a superimposed word "dog" ). When the prediction category class is different from the superimposed one (the S/S and B/B condition), parts of switched labels were the consistent category with superimposed words across different category levels (e.g., a "animal" prediction for a superimposed word "dog" ).

# H  RELATIONSHIP BETWEEN NO-WORD IMAGES AND WORD-SUPERIMPOSED IMAGES

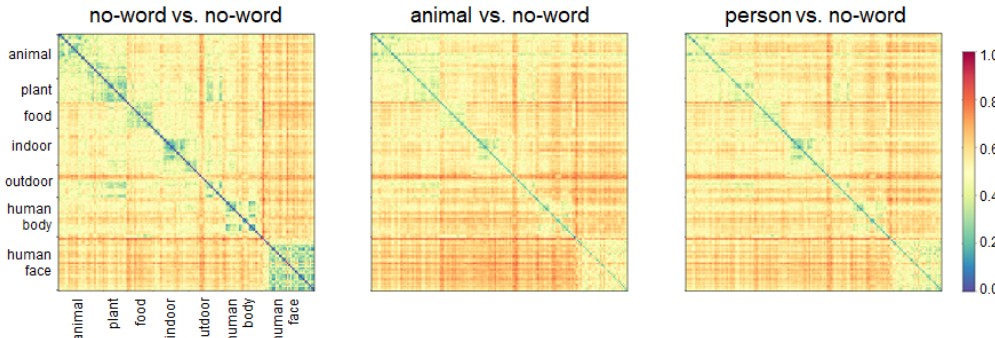

Figure H1: Relationship between no-word and word-superimposed images. We analyzed the representational dissimilarity matrices (RDMs) between no-word images and word-superimposed images for the CLIP image encoder (ViT/32). The vertical axis of each panel shows the no-word images, and the horizontal axis shows the word-superimposed images for the center and right panel. The left panel is the same as the no-word result in Figure 4. If the image encoder has a shared semantic representation for both the superimposed words and the images, the RDMs should show the category-specific similarity when the image category is consistent with the superimposed word. For instance, when the word "animal" is superimposed on the image, the similarity between the word-superimposed images and the animal images should be selectively high. However, results did not show any category-specific similarity, and the RDMs were similar even different words were superimposed. The results are consistent with that the CLIP image encoder has different representations of word forms from visual image representations.

