# OpenReview forum: "Language-biased image classification: evaluation based on semantic representations"
_ICLR.cc/2022/Conference — ICLR 2022 Poster_

### Official Review · Reviewer_P83Y · 2021-10-26

**Correctness:** 1
**Technical Novelty And Significance:** 3
**Empirical Novelty And Significance:** 3
**Recommendation:** 3
**Confidence:** 4

**Main Review:**

The question of what similarity CLIP should assign to text in images
vs.  pictured objects is potentially interesting. There doesn't seem
to be a "right" answer, as both transcribing the red text and naming
the underlying pictured object are arguably justified. The connection
to Rosinski's work is interesting, and understanding how CLIP deals
with the ambiguity of this situation is perhaps interesting from the
perspective of adversarial examples.

I had a few fundamental concerns with this work. First, I didn't
entirely understand the motivation. Goh et al. 2021 demonstrated that
CLIP's predictions might flip if a word was written in the image, but
I hesitate to call this a "misclassification" as the authors say:
given CLIP's objective (matching images/captions), it stands to reason
that, e.g., CLIP would assign a high score to "white dog with the word
electronic written on it", "a white dog", and "electronic" to the
image in the top left of Figure 1. None of these, to me, seem like
outright misclassifications. The authors motivate their consideration
by citing Rosinski's work, in which he showed children pictures of
various objects with correct and incorrect labels, where the children
were specifically tasked with labeling the image (and not the
text). But, CLIP has no access to such asymmetric direction, so to
call transcribing the text incorrect, to me, is misleading.

The authors' most important argument is that CLIP doesn't behave the
same as a human in this setting. The evidence: label-switched
"misclassifications" (where the model predicts the written word rather
than the object depicted) don't depend on the semantic distance to the
distractor word (as measured by word2vec vs. the true object class)
nor the spelling (as measured by Jaro-Winkler), as they do with
humans. The alternate hypothesis would be that, like in humans, it
would be more difficult to name the correct pictured object (say,
"dog") if the distractor were semantically related (e.g., "cat") or
close in spelling (e.g., "bog").

But, my concern regarding the ambiguity of the setup remains: I don't
think it's fair to simply call the text transcription "incorrect". And
so, without accounting for that, I don't think I buy the arguments the
authors are making with respect to comparison to human
experiments. The equivalent human test, IMO, is ill-designed: handing
children images with text on them and asking them to write something
doesn't seem fair, and, for me, neither does this setup.

**Summary Of The Paper:**

The authors investigate the behavior of CLIP when its handed images
superimposed with text. The authors define a label-switched image as
one for which the word being superimposed changes the classification
prediction of the model. They construct a corpus of {12 superordinate}
x {138 basic} x {~150 images with FMRI data} by superimposing labels
over each image and then measuring CLIP's response. The authors
conduct an error analysis of CLIP on these images, arguing that
because CLIP's errors aren't predictable by the semantic or spelling
similarity of the word/pictured object (as they are in a set of human
psychology experiments), CLIP doesn't appear to process images in a
similar fashion to humans.

**Summary Of The Review:**

I think that the authors should continue to peruse this direction,
because I think it's potentially quite interesting: understanding
if/how CLIP can reason about ambiguous scenarios where more than one
answer is correct (in this case: transcription vs. object detection)
is cool. But: because the "correct" answer (or a task description)
isn't ever given to CLIP (as it was to the children in Rosinski's
work), I can't bring myself to be convinced by any of the experimental
claims.  My suggestion to the authors would either to i) consider
fine-tuning CLIP with using supervised data that specifies which of
the tasks should be undertaken (transcription vs. detection); or ii)
consider designing prompts that specify what should be predicted,
e.g., "A photo of the word X written in red font over a picture of Y".

---

> ### Author Response · Authors · 2021-11-16
> **Responses to the reviewer from the authors.**
>
> Thanks a lot for these comments which were very helpful and led us to improve our paper as detailed below.
>
> First, we agree with the point that a label flip due to written words should not be called “misclassification” or “incorrect” because CLIP is trained just for matching. In the revised manuscript, we stopped referring to label flipping as a misclassification or an incorrect error response.
>
> Second, we thank the reviewer for the relevant suggestion to conduct experiments where one uses dedicated prompts to specify what should be answered by the CLIP model. We followed this idea and conducted additional experiments which we included in the revised manuscript (pages 8-9 in the main text and page 17 in Appendix D).
>
> First, we created three prompts that specify that the image content is what should be predicted by the CLIP model (red in Table 2) and three prompts that specify that the superimposed word is what should be predicted by the CLIP model  (blue in Table 2). We used each prompt for all word-superimposed images. These prompts did not change according to what superimposed word was presented. For instance, we used the same prompt, "a red word label over a picture of a X" for all images. In addition to such fixed prompts, we created a prompt that varies according to the superimposed word on each image. We used a prompt "a photo of the word Y written in a red font over a picture of a X," where X is the label to be answered by the model and Y is the same variable word as superimposed on each image (orange in Table D1).  For instance, we used a prompt "a photo of the word electric in a red font over a picture of a X" when the superimposed word is "electric," and a prompt "a photo of the word dog in a red font over a picture of a X" when the superimposed word is "dog."
>
> The new results of the additional experiments confirmed our main findings of language-biased image classification. More specifically, the results showed that the label switching changed with the different prompts, i.e., the focus of the task (image content or superimposed word). When the prompt was focused on the image content (red in Table 2) the switching rates were lower than when it was focused on the superimposed word (blue in Table 2). However, the language-biased classification was high for all conditions. The results on the variable prompt (orange in Table 2) showed that the label switching rate for the S/S and B/B conditions decreased more than the prompt conditions that specify the image content (red in Table 2). The finding suggests that describing the superimposed word in the prompt is effective in suppressing the language-biased classification. Even so, the label-switching rates were still high, more than 30% for all conditions. In addition, we analyzed the semantic similarity between the image and superimposed words for label-switched and unswitched images. The results showed that, in any prompts, the semantic relationship did not explain the label-switching effect.
>
> Based on these additional experiments, we acknowledge the importance of task objective dependency but observe that the dependency does not overturn our main findings.
>
> Our motivation is not to create the exact same condition as the human experiment. Rather, we focus on the methodological strategy that controlling the semantic relationship between images and superimposed words can reveal the underlying functional mechanism. In fact, the semantic dependency in human picture-word interference has been shown in various experimental paradigms. We believe that our CLIP investigation and the additional experiments also contribute to understanding the functional mechanisms.

---

> > ### Author Response · Authors · 2021-11-19
> > **Reaction to our response**
> >
> > The authors thank again the reviewer for his/her time and valuable review. In addition, we would like to highlight that the discussion period will end Monday.
> >
> > We hope that we have covered your concerns. It would be helpful to know your reaction to our response while there is still time to engage in discussion.

---

> > > ### Comment · Reviewer_P83Y · 2021-11-22
> > > **thanks for the updates!**
> > >
> > > Just replying to say: thank you for the updates ! I have seen them and I think the prompts you wrote do provide a zero-shot experiment that doesn't suffer from the same symmetry problems as the unprompted cases. I will take these updates into account in my reviewing capacity for this work going forward.

---

### Official Review · Reviewer_9yiw · 2021-10-30

**Correctness:** 3
**Technical Novelty And Significance:** 3
**Empirical Novelty And Significance:** 4
**Recommendation:** 8
**Confidence:** 4

**Main Review:**

[Overall] This is a very interesting piece of work that analyzes whether the state-of-the-art artificial vision and language model; CLIP, resembles human cognition or not.

[Formatting]
- Figures and Tables should be aligned to the top of the page.

[Writing]
- I though that CLIP is trained by inputting image and words separately, but according to the authors statement in Section 5: "how CLIP acquired the ability to read superimposed words", it seems that it models language information from the image (superimposed text). Is this true? If so, I doubt it, and accordingly, results and discussions based on it may be results of some undiscussed reasons. Please also introduce in detail the task settings so that this kind of confusion will not occur.

**Summary Of The Paper:**

The authors introduce a framework to evaluate the picture-word interference in CLIP which is known to exist in human cognition.
It is designed to test whether language-biased decisions occur across different category levels, and the extent to which picture-word interference in CLIP depends on the semantic similarity between superimposed words and images. Experimental results show that presenting words disturbed the CLIP image classification even across different category levels, the effect did not depend on the semantic relationship between images and words, and the superimposed word representation in the CLIP image encoder is not shared with the image representation.

**Summary Of The Review:**

Work like this should is very important to understand how and why the state-of-the-art models perform well, what their limitations are, and how we can improve them.

---

> ### Comment · Reviewer_9yiw · 2021-11-19
> **My concern**
>
> Dear authors, can you comment on my concern in [Writing]?

---

> > ### Author Response · Authors · 2021-11-19
> > **Response to Reviewer 9yiw**
> >
> > We sincerely apologize for the delay in getting back to you. We were revising the main text according to your comments. We will update the revised manuscript soon but will answer your concerns first.
> >
> > > I though that CLIP is trained by inputting image and words separately, but according to the authors statement in Section 5: "how CLIP acquired the ability to read superimposed words", it seems that it models language information from the image (superimposed text). Is this true? If so, I doubt it, and accordingly, results and discussions based on it may be results of some undiscussed reasons. Please also introduce in detail the task settings so that this kind of confusion will not occur.
> >
> > We thank the reviewer for pointing out our unclear description. The CLIP image and text encoders separately input the images and texts in training, as you described. These encoders are trained to match representations of images and texts. We agree that we should not say "the ability to read superimposed words" because the description is confusing. The image encoder only visually recognizes the written word and matches the representation with the text encoder's one.
> >
> > > Figures and Tables should be aligned to the top of the page.
> >
> > We aligned these figures and tables to the top of the page in the current manuscript.

---

> > > ### Comment · Reviewer_9yiw · 2021-11-19
> > > **Thanks!**
> > >
> > > for the response.
> > > That cleared my concern. Please revise the manuscript accordingly.
> > > Otherwise, I think this is a very interesting piece of work!!!

---

> > > > ### Author Response · Authors · 2021-11-20
> > > > **Upload the revised pdf**
> > > >
> > > > We appreciate your response. We uploaded the revised pdf now. We also uploaded a supplementary pdf file ("iclr2022_3_highlighted.pdf") in which the differences from the original manuscript are highlighted by using colored text. We hope that we have covered your concerns.

---

### Official Review · Reviewer_WKSS · 2021-11-02

**Correctness:** 3
**Technical Novelty And Significance:** 3
**Empirical Novelty And Significance:** 3
**Recommendation:** 6
**Confidence:** 3

**Main Review:**

Overall this paper is well-written. The motivation and background are clearly stated. To evaluate the effect of word-picture interference, the authors applied multiple analyses, including assessing the model behaviors under different conditions (section 4.2, 4,5), connecting model behavior to label similarity (section 4.3, 4.4), and comparing the similarity of visual model representations (section 4.6). The methods are described in details and should be easy for readers to reproduce their analyses. The authors are trying to figure out how the visual representations of word form and object images interfere with each other and bias the classification behavior. This is an intriguing question driven by observations and theories from cognitive science.

Comments:
1. In general, I agree with the authors' claims. However, I feel there exists some interesting findings in the current results remain further explanation or investigation. First, during pre-training, the CLIP model already matches the representation of "*object in visual form*" and "*word in textual form*" in the same metric space. And the authors further find that the image classification (section 4.2, 4.5) is strongly biased by the superimposed word (e.g., examples in Figure A1), even in the B/S or S/B condition (e.g., recognize tulip image labeled 'electronic' to semantic class 'keyboard', or recognize tulip image labeled 'laptop' to semantic class 'electronic'). Does it imply that the representational space of "*words in visual form*" should have similar structure (in the sense of pair-wise similarity) as the representational space of "*word in textual form*"? However, all the similarity analyses based on representations (section 4.3, 4.4, 4.6) showed non-distinguishable results across conditions, which lead to the authors' statement that "the CLIP image encoder has different representations of word forms from visual image representations". But it seems that at least both "word forms" and "object image" representations from CLIP image encoder captures semantic similarity in the text-based word space. What's authors interpretation on this?

2. I am a bit surprised about the good performance of CLIP recognizing spelling word into the same category word. Were all the results based on pre-trained CLIP without fine-tuning? Was CLIP trained on some images with visual word forms to learn typographic features?

3. Were all 'misclassified' word-embedded images classified as the superimposed label? If not, it might be better to divide the 'misclassification' condition into two cases: 'biased' (predict the superimposed label) and 'random' (predict a label not original or superimposed), just to make results in section 4.2 more informative.

4. The left box in each panel of Figure 4 is a bit ambiguous to me. What is 'label-switched' for 'original images'?

5. What would be the model behavior and similarity results if the image is interfered with a nonsense pseudoword?

6. Typo: 'fisrt' should be 'first' on page 5.

7. What does the "semantic compositionality" in the paper title refer to? I feel this term is weakly connected to the main text.

**Summary Of The Paper:**

This paper introduced a benchmark task for evaluating how semantic processing of visual word form interfere with the recognition processing of an object image in word-embedded images. By placing category word (e.g., superordinate category like 'electronic' or basic category like 'laptop') in the center of an image, the authors evaluated the 1) the misclassification rates of CLIP; 2) the semantic/spelling similarity between original and superimposed labels under different conditions; 3) the change of model prediction probability; 4) RSA on images with different types of picture-word interference.
In general, this work provided interesting new insights on how language information bias the image classification in visual-language models like CLIP. The authors concluded that CLIP can recognize the visual word form as a word but fails to encode word form and object image that share the same category (e.g. word 'dog' and picture 'dog') with similar visual representations.

**Summary Of The Review:**

This paper introduced a benchmark task for evaluating picture-word interference in visual-language joint learning models. The benchmark was tested with the latest model CLIP. The evaluation results brought novel insights to understand the recognition process of the CLIP image encoder, showing its distinction on visual representations of word forms and object images. This provides a new perspective for future studies to evaluate the interference of multi-modal processing in AI models. Some results and figures may need further explanation or clarification for better understanding of their findings.

---

> ### Author Response · Authors · 2021-11-19
> **Response to Reviewer WKSS (1/2)**
>
> We thank the reviewer to take the time and give us helpful comments and suggestions.
> We have revised the manuscript on the basis of your comments and suggestions as follows. The revised pdf will be posted soon.
>
> > 1. In general, I agree with the authors' claims. However, I feel there exists some interesting findings in the current results remain further explanation or investigation. First, during pre-training, the CLIP model already matches the representation of "object in visual form" and "word in textual form" in the same metric space. And the authors further find that the image classification (section 4.2, 4.5) is strongly biased by the superimposed word (e.g., examples in Figure A1), even in the B/S or S/B condition (e.g., recognize tulip image labeled 'electronic' to semantic class 'keyboard', or recognize tulip image labeled 'laptop' to semantic class 'electronic'). Does it imply that the representational space of "words in visual form" should have similar structure (in the sense of pair-wise similarity) as the representational space of "word in textual form"? However, all the similarity analyses based on representations (section 4.3, 4.4, 4.6) showed non-distinguishable results across conditions, which lead to the authors' statement that "the CLIP image encoder has different representations of word forms from visual image representations". But it seems that at least both "word forms" and "object image" representations from CLIP image encoder captures semantic similarity in the text-based word space. What's authors interpretation on this?
>
> [Response] As the reviewer mentioned, our results suggest that "word forms" and "object image" representations are separated in the image encoder, but both of them capture semantic similarity in the text-based word space. Also, the matching is strongly biased to word forms. We speculate that this is because the word space dimension by the text encoder is high enough to match with both of the separate representations, and the CLIP acquires the matching in training. If the CLIP training data includes many images with word forms, a bias to word forms could emerge when various images of a specific category (e.g., various pizza images) include the same word (e.g., written word "pizza"). We do not have conclusive evidence for this speculation at the moment. However, by conducting an additional experiments (page 20, Appendix H), we confirmed whether our current finding was robust and whether this untangled state was the problem to be focused on in the future. Specifically, we analyzed the representational dissimilarity matrices (RDMs) between no-word images and word-superimposed images for the CLIP image encoder. If the image encoder has a shared semantic representation for both the superimposed words and the images, the RDMs should show the category-specific similarity when the image category is consistent with the superimposed word. For instance, when the word "animal" is superimposed on the image, the similarity between the word-superimposed images and the animal images should be selectively high. However, results did not show any category-specific similarity, and the RDMs were similar even different words were superimposed. The result confirmed that our finding, i.e., "the CLIP image encoder has different representations of word forms from visual image representations," is robust.
>
>
>
> > 2. I am a bit surprised about the good performance of CLIP recognizing spelling word into the same category word. Were all the results based on pre-trained CLIP without fine-tuning? Was CLIP trained on some images with visual word forms to learn typographic features?
>
> [Response] All the results were based on pre-trained CLIP without fine-tuning. There is a possibility that the CLIP training data includes some written words like a signboard in an outdoor scene. In the revised manuscript, we conducted an additional experiment where one uses dedicated prompts to specify what should be answered by the CLIP model (pages 8-9, Table 2). Results showed that a bias to word forms was increased when the prompt was focused on the superimposed word.

---

> > ### Author Response · Authors · 2021-11-19
> > **Response to Reviewer WKSS (2/2)**
> >
> > > 3. Were all 'misclassified' word-embedded images classified as the superimposed label? If not, it might be better to divide the 'misclassification' condition into two cases: 'biased' (predict the superimposed label) and 'random' (predict a label not original or superimposed), just to make results in section 4.2 more informative.
> >
> > [Response] We additionally analyzed the relationship between switched labels and superimposed words. When the prediction category class is consistent with the superimposed word class (the S/S and B/B condition), most switched labels were the same word as superimposed on the images (more than 90%). When the prediction category class is inconsistent with the superimposed one (the S/S and B/B condition), 30 ~ 50 % of switched labels were the same category with superimposed words across different category levels. We described how the label-switched images were distributed in the revised manuscript (Appendix G).
> >
> > > 4. The left box in each panel of Figure 4 is a bit ambiguous to me. What is 'label-switched' for 'original images'?
> >
> > [Response] We revised Figure 4’s legend and description. The original prediction means the probability for no-word images. In the figure, the original prediction is categorized into label-switched or unswitched conditions. The label-switched and unswitched conditions refer to if the original prediction is switched in the new prediction for word-superimposed images.  Now this section is moved to Appendix E because we added the additional experiment in the main text.
> >
> > > 5. What would be the model behavior and similarity results if the image is interfered with a nonsense pseudoword?
> >
> > [Response] We conducted an additional experiment using nonsense pseudowords and random words. We created two pseudowords from the word "animal" by a multilingual pseudoword generator, Wuggy (Keuleers & Brysbaert, 2010). We also used one random word used in a previous human study. Results showed that the label switching rate decreased for these words (Appendix F). We also analyzed the RDM for the images on which these words were superimposed. Our results showed that when nonsense pseudo and random words were superimposed on the images, the effect of words on the representation was weaker than meaningful words. The finding suggests that presenting meaningful words on the images can be a factor affecting the representation of the image encoder.
> >
> > > 6. Typo: 'fisrt' should be 'first' on page 5.
> >
> > [Response] We corrected the typo.
> >
> > > 7. What does the "semantic compositionality" in the paper title refer to? I feel this term is weakly connected to the main text.
> >
> > [Response] We used this term because our benchmark test covers hierarchical cross-category evaluation. However, indeed, this looks like not cover the paper's scope (e.g., semantic relation between images and words). We changed it to semantic representations.

---

> > > ### Author Response · Authors · 2021-11-20
> > > **Upload the revised pdf**
> > >
> > > We uploaded the revised pdf now. We also uploaded a supplementary pdf file ("iclr2022_3_highlighted.pdf") in which the differences from the original manuscript are highlighted by using colored text. We hope that we have covered your concerns.

---

### Official Review · Reviewer_gSM4 · 2021-11-02

**Correctness:** 3
**Technical Novelty And Significance:** 3
**Empirical Novelty And Significance:** 2
**Recommendation:** 6
**Confidence:** 3

**Main Review:**

Strengths:
- The resulting dataset provides a useful tool to evaluate the bias of image recognition models that are jointly trained with language. This can be used by future researchers to evaluate whether their models acquire human-like biases.
- The authors present different analyses to disentangle the effects of picture-word interference in CLIP.

Weaknesses:
- The writing can be improved. I found hard to follow some of the results sections. Figure 1 could also be improved to depict an example of the task itself (Fig A1 is already better), which is unclear solely by the figure and potentially confuses the reader. Similarly, Figure 2 could be improved by showing more outputs, those corresponding to the target behaviour and those corresponding to the interference behaviour.
- The RSA analysis shows that CNNs are less affected by superimposed words than CLIP. However, as also mentioned by the authors in Section 5, CLIP was trained on a different dataset. It would have been instructive to train CLIP on ImageNet and investigate whether the language-biased modelling is still different from CNNs. It is, in fact, unclear to me whether CNNs might observe similar patterns when trained on large, noisy data.

**Summary Of The Paper:**

This paper proposes a testbed for picture-word interference in image classification models. The authors specifically investigate the performance of CLIP to understand whether such language-biased model shows similar interferences to the ones observed in humans. The dataset consists of images superimposed with words, representing two different category levels (basic and superordinate). The experiments on CLIP show that the model is affected by superimposed words, but independently of their semantic relationship with the underlying image. Further analysis shows that CLIP image representations are different from the ones of superimposed words, while this does not happen in two ImageNet-based CNNs.

**Summary Of The Review:**

The paper presents a useful dataset to evaluate word-superimposed image recognition models. The analysis on CLIP shows how it behaves for different categories of superimposed words. The writing can be improved, especially around the presentation of the task and some of the results. Further experiments on ImageNet-based CLIP would make the paper stronger.

---

> ### Author Response · Authors · 2021-11-19
> **Response to Reviewer gSM4**
>
> We thank the reviewer to take the time and give us helpful comments and suggestions. We have revised the manuscript on the basis of your comments and suggestions as follows. The revised pdf will be posted soon.
>
>
>
> > The RSA analysis shows that CNNs are less affected by superimposed words than CLIP. However, as also mentioned by the authors in Section 5, CLIP was trained on a different dataset. It would have been instructive to train CLIP on ImageNet and investigate whether the language-biased modelling is still different from CNNs. It is, in fact, unclear to me whether CNNs might observe similar patterns when trained on large, noisy data.
>
>
> [Response]
> We agree that it is unfair to compare CNNs with CLIP because they are trained using different datasets. Our purpose of the comparison was to confirm whether our observation, i.e., the CLIP representation is affected by superimposed words, is due to an artifact not related to words. For that purpose, we think that we should explore the current CLIP more in detail rather than investigate the CLIP trained by different datasets. In the revised manuscript, to confirm whether our findings are robust, we present new additional analyses of the representation of the CLIP image encoder.
>
> We conducted two analyses, and both were consistent with our findings. First, we analyzed the representational dissimilarity matrices (RDMs) between no-word images and word-superimposed images for the CLIP image encoder. If the image encoder has a shared semantic representation for both the superimposed words and the images, the RDMs should show the category-specific similarity when the image category is consistent with the superimposed word. For instance, when the word "animal" is superimposed on the image, the similarity between the word-superimposed images and the animal images should be selectively high. However, results did not show any category-specific similarity, and the RDMs were similar even when different words were superimposed.
>
> Next, we analyzed the RDM for the images on which nonsense pseudo words were superimposed. Our results showed that when nonsense pseudo words were superimposed on the images, the effect of words on the representation was weaker than meaningful words. The finding suggests that presenting meaningful (nonrandom) words on the images can be a factor affecting the representation of the image encoder.
>
> > The writing can be improved. I found hard to follow some of the results sections. Figure 1 could also be improved to depict an example of the task itself (Fig A1 is already better), which is unclear solely by the figure and potentially confuses the reader. Similarly, Figure 2 could be improved by showing more outputs, those corresponding to the target behaviour and those corresponding to the interference behaviour.
>
> [Response] We modified Figures 1 & 2 so that they include clear task examples. Also, we have clarified the writing in the revised manuscript, especially in the results section.

---

> > ### Author Response · Authors · 2021-11-20
> > **Upload the revised pdf**
> >
> > We uploaded the revised pdf now. We also uploaded a supplementary pdf file ("iclr2022_3_highlighted.pdf") in which the differences from the original manuscript are highlighted by using colored text. We hope that we have covered your concerns.

---

### Decision · Program_Chairs · 2022-01-20

**Decision:**

Accept (Poster)

**Comment:**

This paper presents a new benchmark task for models similar to CLIP for evaluating how visual word forms interfere with the visual recognition of objects in images when the former are superimposed on the latter ones. Specifically, by superimposing words belonging to different categories  (e.g., hypernyms vs basic labels) the authors study the misclassification rates of CLIP under different degrees of varying similarity between the original and superimposes labels.

All reviewers agreed that this is a novel and interesting study which, by productively using insights from cognitive science literature on language biases, aims at shedding light on the inner-workings of a popular artificial model. The main concern raised by reviewer P83Y was regarding the claims around misclassification rates. Indeed, since CLIP was not taught (e.g., by fune-tuning or few-shot prompting) which of the two labels (i.e., the written or the visual) is the correct one, it's not fair to assess its performance on this way. While this is strictly true, the experimental protocols presented in Sections 4.3/4/5 are still a valid way to assess representational inference. Moreover, the authors have followed P83Y suggestions and incorporated a few-shot prompting experiment in Section 4.6.

All in all, I think this will make for an addition to the ICLR program and thus I'm recommending accepting this paper.

(Minor comment: WKSS rightly pointing that this paper has, at best, a loose connection to compositionality. The authors changed compositionality -> representations which is a better fit, so please make sure to change the title also in Openreview when prompted.)